# Comparing the Osteogenic Potential and Bone Regeneration Capacities of Dedifferentiated Fat Cells and Adipose-Derived Stem Cells In Vitro and In Vivo: Application of DFAT Cells Isolated by a Mesh Method

**DOI:** 10.3390/ijms222212392

**Published:** 2021-11-17

**Authors:** Kiyofumi Takabatake, Masakazu Matsubara, Eiki Yamachika, Yuki Fujita, Yuki Arimura, Kazuki Nakatsuji, Keisuke Nakano, Histoshi Nagatsuka, Seiji Iida

**Affiliations:** 1Department of Oral Pathology and Medicine, Graduate School of Medicine, Dentistry and Pharmaceutical Sciences, Okayama University, Okayama 700-8525, Japan; gmd422094@s.okayama-u.ac.jp (K.T.); pir19btp@okayama-u.ac.jp (K.N.); jin@okayama-u.ac.jp (H.N.); 2Department of Oral and Maxillofacial Reconstructive Surgery, Graduate School of Medicine, Dentistry and Pharmaceutical Sciences, Okayama University, Okayama 700-8525, Japan; pprd1898@okayama-u.ac.jp (Y.A.); de18043@s.okayama-u.ac.jp (K.N.); iida-s1@cc.okayama-u.ac.jp (S.I.); 3Department of Dentistry, National Hospital Organization Okayama Medical Center, Okayama 701-1192, Japan; 4Department of Oral and Maxillofacial Reconstructive Surgery, Okayama University Hospital, Okayama 700-8525, Japan; yfujita@okayama-u.ac.jp

**Keywords:** dedifferentiated fat cells (DFAT cells), adipose-derived stem cells (ASCs), bone regeneration, mesh culture method

## Abstract

Background: We investigated and compared the osteogenic potential and bone regeneration capacities of dedifferentiated fat cells (DFAT cells) and adipose-derived stem cells (ASCs). Method: We isolated DFAT cells and ASCs from GFP mice. DFAT cells were established by a new culture method using a mesh culture instead of a ceiling culture. The isolated DFAT cells and ASCs were incubated in osteogenic medium, then alizarin red staining, alkaline phosphatase (ALP) assays, and RT-PCR (for RUNX2, osteopontin, DLX5, osterix, and osteocalcin) were performed to evaluate the osteoblastic differentiation ability of both cell types in vitro. In vivo, the DFAT cells and ASCs were incubated in osteogenic medium for four weeks and seeded on collagen composite scaffolds, then implanted subcutaneously into the backs of mice. We then performed hematoxylin and eosin staining and immunostaining for GFP and osteocalcin. Results: The alizarin red-stained areas in DFAT cells showed weak calcification ability at two weeks, but high calcification ability at three weeks, similar to ASCs. The ALP levels of ASCs increased earlier than in DFAT cells and showed a significant difference (*p* < 0.05) at 6 and 9 days. The ALP levels of DFATs were higher than those of ASCs after 12 days. The expression levels of osteoblast marker genes (osterix and osteocalcin) of DFAT cells and ASCs were higher after osteogenic differentiation culture. Conclusion: DFAT cells are easily isolated from a small amount of adipose tissue and are readily expanded with high purity; thus, DFAT cells are applicable to many tissue-engineering strategies and cell-based therapies.

## 1. Introduction

Cell-based tissue-engineering approaches are potential therapeutic strategies for bone repair and bone regeneration. Identifying an optimal cell source for generating functional osteoblasts is critical to achieve clinical success with bone regeneration strategies. Cells for tissue engineering in bone regeneration ideally need to possess pluripotency, high cell proliferation ability, and high purity. Recently, various cell types for tissue regeneration have been identified, but the proper selection of the cell source is still important for clinical applications.

Mesenchymal stem cells (MSCs) are multipotent somatic stem cells that can differentiate into a variety of cell types such as osteoblasts, chondrocytes, adipocytes, and myocytes [1,2,3,4]. MSCs were originally isolated from bone marrow, adipose tissue, periosteum, synovium, and dental pulp [5,6].

Bone marrow mesenchymal stem cells (BMDCs) are readily available and possess high osteogenic capacity; as such, they are considered an appropriate stem cell populations for bone regeneration. Thus, they are widely used for efficiency comparisons with other cell sources [7,8,9,10,11,12]. BMDCs can be collected from bone marrow aspirate, but the method of harvesting BMDCs is invasive. Although there are advantages to their use, these cells have limitations.

Adipose-derived stem cells (ASCs) show a multilineage potential similar to BMDCs and can be easily harvested from mature adipose tissue. ASCs can differentiate along multiple lineages, resulting in adipocytes, osteoblasts, chondrocytes, myocytes, endothelial cells, and hepatocytes [13,14,15,16,17]. In addition, ASCs are readily available in large quantities with minimal morbidity and discomfort associated with their harvest [13,14,15,16]. ASCs are extensively used for bone tissue engineering, and their utility has been reported in various studies [17,18,19,20,21].

However, ASCs are a heterogeneous cell population because they are obtained from the non-adipocyte fraction in adipose tissue, including the stromal vascular fraction (SVF) [13,14,15]. In addition, ASCs at early passages include contaminating endothelial and smooth muscle cells and pericytes [13]. Therefore, for these stem cells to be widely used in clinical applications, cell sources with high purity are needed.

Mature adipocytes are the most abundant cell type in adipose tissue [22]. Mature adipocytes that contain a large single lipid droplet are generally considered to be stable cells that have reached the terminal stage of differentiation, and these cells are considered to have already lost their proliferative ability. Yagi et al. established a preadipocyte cell line derived from mature adipocytes of ddY mice and designated these cells as dedifferentiated fat (DFAT) cells [23]. They reported that DFAT cells have high proliferative activity and, similar to BMDCs, have the potential to differentiate into mesenchymal tissue lineages [22].

Some previous research has found that DFAT cells generated in specific culture conditions can differentiate into adipocytes, osteoblasts, chondrocytes, myofibroblasts, and cardiomyocytes [20,24,25,26]. Transplantation of DFAT cells into injured tissue contributes to the regeneration of damaged tissues, including of the bladder, urethra, heart, and spinal cord [24,26,27]. Because adipose tissue, which is the source of DFAT cells, can be collected from small amounts of subcutaneous fat and can be generated from anyone regardless of donor age, DFAT cells have potential applications in cell-based therapies for a variety of diseases, including metabolic bone disorders, such as osteoporosis, that commonly affect elderly subjects.

Previous methods of generating DFAT cells have not included adipocytes that do not attach to plastic surfaces [22,28,29,30,31]. In fact, DFAT cell used for tissue engineering are now cultured using ceiling culture methods. However, we have devised a mesh method that uses floating mature fat cells for a more efficient collection method.

The purpose of the present study was to evaluate the bone differentiation ability of DFAT cells collected by our newly developed mesh method. In addition, we compared the osteoblastic differentiation abilities of DFAT cells and ASCs in vitro and in vivo.

## 2. Results

### 2.1. Isolation of DFAT Cells and ASCs from Adipose Tissue

The mesh culture method is illustrated in Figure 1. In this method, DFAT cells are isolated from small pieces of subcutaneous adipose tissue that include a large single lipid droplet and washed repeatedly with phosphate-buffered saline (PBS) until the washes are clear. Approximately 1 g of adipose tissue was isolated from the subcutaneous fat of GFP mice and digested in a solution of 1 mg/mL collagenase type II and 1 mg/mL dispase for 2 h at 37 °C. After filtration through 100-μm nylon filters and centrifugation (135× *g*, 10 min), we collected the floating top layer of adipocytes for DFAT cells and the settled cell pellet for SVFs. The cultured adipocytes were added to control medium plates fitted with 40-μm meshes and incubated for five days. DFAT cells generated from adipocytes passed through the mesh and attached to the bottom of the dishes. After five days, the meshes with remaining adipocytes were removed. This method of collecting DFAT cells did not include the attachment of the adipocytes to plastic surfaces or ceiling culture.

In DFAT cell culture, mature adipocytes divided asymmetrically into large lipid-filled adipocytes and small daughter cells without lipids, and the new lipid-free cells proliferated. The new lipid-free cells showed a fibroblast-like shape. In ASC culture, there was no lipid content, and ASCs also exhibited a fibroblast-like morphology (Figure 2).

### 2.2. Osteoblastic Differentiation of DFAT Cells and ASCs In Vitro

#### 2.2.1. Comparison of Calcification Ability by Alizarin Red Staining

Alizarin red staining was used to quantify the mineral matrix depositions of DFAT cells and ASCs after 1, 2, and 3 weeks of osteogenic induction culture. The mineral matrix deposition of the ASCs was found to be significantly higher than that of the DFAT cells up to 2 weeks. However, after 3 weeks of culture, no obvious difference was observed between the DFAT cell group and the ASC group (Figure 3A).

#### 2.2.2. Comparison of Alkaline Phosphatase (ALP) Activity

Protein levels of ALP were measured by ELISA in DFAT cells and ASCs after exposure to osteogenic medium. ALP activity of the ASCs was significantly higher than that of the DFAT cells at 6 and 9 days. However, after 12 days, DFAT cells displayed stronger ALP activity than ASCs in vivo during the late differentiation period (Figure 3B).

#### 2.2.3. Reverse Transcription-Polymerase Chain Reaction (RT-PCR) Analysis

The expression of osteoblastic marker genes of DFAT cells and ASCs was analyzed by RT-PCR before (0 weeks) and after osteogenic differentiation (1–3 weeks). RUNX2 and osteopontin were higher in ASCs than in DFAT cells at 2 and 3 weeks, but at 3 weeks they were at the same level. DLX5 was markedly higher for ASCs than for DFAT cells at 1, 2, and 3 weeks. Osterix was the same between DFAT cells and ASCs; however, osteocalcin was higher in DFAT cells than in ASCs at 2 and 3 weeks (Figure 4).

### 2.3. Formation of Osteoid Tissue by Transplanted DFAT Cells and ASCs

We examined whether DFAT cells and ASCs could form osteoid tissue in vivo. At 4 weeks after implantation, HE staining and histological analyses were performed. At low magnification, new bone formation was observed in both the DFAT cell group and ASC group. At high magnification, lamellar bone was readily observed in all samples in both groups. In all samples of both groups, osteoblasts lining the bone matrix and numerous osteocytes incorporated in the lacunae of the newly generated bone structure were observed. In addition, osteocalcin- and GFP-positive cells were present in bone tissue in both groups. Thus, newly formed bone tissue was derived from implanted cells (Figure 5).

## 3. Discussion

To the best of our knowledge, the present study is the first attempt using a mesh method to evaluate the osteogenic differentiation abilities of DFAT cells in vitro and to investigate new bone formation efficiency by comparing them with ASCs. Overall, the in vitro results showed that these DFAT cells possess high osteogenic differentiation potential, similar to that of ASCs. In addition, the in vivo results showed that the bone regenerative capacity of the DFAT cells is similar to that of the ASCs.

ASCs have been already widely investigated as a cellular source of potential MSCs in regenerative medicine. In previous studies, DFAT cells showed better MSC properties than ASCs generated from the same adipose tissue [32,33]. In addition, compared with ASCs, DFAT cells are a more homogeneous cell population [22]. Furthermore, these studies indicated that the osteoblastic differentiation potential was greater for DFAT cells than for ASCs [32,34]. DFAT cells have more than 99.5% homology with ASCs in comprehensive gene expression analysis, and the secretion profile of humoral factors of DFAT cells is similar to that of ASCs, but HGF, VEGF, SDF-1, and leptin secretion tends to be high, and there is a stable and high angiogenesis effect in DFAT cells. Accordingly, some studies have reported that DFAT cells are a homogeneous cell population because they are isolated due to the buoyancy of adipocytes in ceiling culture [14,22,35,36]. In contrast, ASCs are a heterogeneous cell population because they are isolated from the SVF, which contains various cell types without mature adipocytes.

In the present study, the cell calcification ability, osteogenic differentiation potential, and osteogenic gene expression of DFAT cells and ASCs were compared by in vitro assays. Our findings revealed that both cell types had osteogenic differentiation properties; however, the in vitro results showed that ASCs possessed higher and earlier osteogenic differentiation potential than DFAT cells (Figure 3). In addition, ASCs showed higher expression of early osteoblast marker genes than DFAT cells (Figure 4). Other studies have indicated that DFAT cells have a higher proliferation rate, availability, and cell number than ASCs [22,34]. The cell population heterogeneity could be induced by local variations in cell density, local buildup of extracellular matrix, and be amplified as the cells grow. Additionally, cell growth and local cell death may trigger some cell differentiation and the loss of all or some pluripotency markers. Therefore, differences between DFAT cells and ASCs in their rate of differentiation into bone tissue and differences in expression markers are thought to be due to the composition of these cell populations. Additionally, we introduced additional steps to enhance DFAT cell preparation, in which DFAT cells were separated from the floating adipocytes and passed through a mesh on the way to the dish bottom. In this way, we addressed disadvantages such as the potential attachment of contaminating cells to the ceiling [22,32,33,34,35]. This mesh method, and the separation of DFAT cells from adipocytes during preparation improved the homogeneity of DFAT cells.

Our results in vitro suggest that DFAT cells collected by the mesh method used in this study may be more mature than DFAT cells collected in ceiling culture, and DFAT cells collected by the mesh method may be a homogenous population with a more defined tendency for osteogenic differentiation rather than multilineage potential. However, there may be some possible limitations in investigation of DFAT osteogenic differentiation of this study. To examine the biological character of DFAT cells in more detail, it is necessary to investigate the surface markers using fluorescence-activated cell sorting.

In addition, our results in vivo showed osteoid formation in the removed implants of both DFAT and ASC groups, and that osteocalcin and GFP were positive in both groups. These results suggest that our newly generated DFAT cells have the same osteogenic potential in vivo as ASCs and are a cell source that can be effectively applied clinically.

## 4. Materials and Methods

### 4.1. Isolation and Culture of DFAT Cells and ASCs

Approximately 1 g of adipose tissue was isolated from the subcutaneous fat of GFP mice and digested in a solution of 1 mg/mL collagenase type II and 1 mg/mL dispase for 2 h at 37 °C. After filtration through 100-μm nylon filters and centrifugation (135× *g*, 10 min), we collected the floating top layer of adipocytes as DFAT cells and the settled cell pellet as SVFs. The cultured adipocytes were added to control medium plates fitted with 40-μm meshes (BD, Franklin Lakes, NJ, USA) and incubated for five days (Figure 1).

### 4.2. Mineralization Assay by Alizarin Red Staining

To assess osteogenic differentiation, DFAT cells and ASCs were cultured in 12-well plates for 1, 2, and 3 weeks in osteogenic medium consisting of DMEM, 10% FBS, 10 mM β-glycerophosphate (Sigma-Aldrich, St. Louis, MO, USA), and 0.05 mM L-ascorbic acid (Sigma-Aldrich, St. Louis, MO, USA). The cultured cells were washed once with PBS and fixed with 95% ethanol at 37 °C for 15 min. The fixed cells were washed with distilled water and subsequently stained for 15 min with 1% alizarin red S (Katayama chemical Industries, Co. Ltd., Osaka, Japan) solution.

### 4.3. ELISA for Quantification of Alkaline Phosphatase

After reaching confluence, the culture medium was replaced by osteogenic medium. Alkaline phosphatase activity was measured by the p-Nitrophenyl Phosphatase Substrate method (FUJIFILM Wako Pure Chemical Corporation, Osaka, Japan) according to the manufacturer’s instructions at 0, 3, 6, 9, 12, and 15 days.

### 4.4. Immunohistochemical Staining of GFP and Osteocalcin

In this study, rabbit polyclonal anti-GFP antibody (MBL, Nagoya, Japan) and rabbit polyclonal anti-osteocalcin (Santa Cruz Biotechnology, Inc., Dallas, TX, USA) were used. The sections were deparaffinized in a series of xylene solutions for 15 min and then rehydrated, and incubated in 0.1% trypsin (Difco Laboratories, Detroit, MI, USA) for 5 min at 37 °C. Immunohistochemistry was performed using anti-GFP polyclonal rabbit antibody at a dilution of 1:1000 or anti-osteocalcin polyclonal rabbit antibody at a dilution of 1:100 for 120 min at room temperature. The tagging of primary antibody was achieved by subsequent application of anti-goat antibody using the Histofine SAB-Po^®^ Kit (Nichirei, Tokyo, Japan) following the instructions of the manufacturer. Immunoreactivity was visualized using diaminobenzidine (DAB)/H_2_O_2_ solution (Histofine DAB substrate; Nichirei, Tokyo, Japan), and the sections were counterstained with Mayer’s hematoxylin.

### 4.5. Transplantation and Histological Evaluation

All animal experiments were performed in accordance with relevant guidelines and regulations and were approved by the institutional committees at Okayama University (OKU-2015118, 18 April 2015). BALB-c nu-nu mice were subjected to intramuscular anesthesia with ketamine (Fuji Chemical Industry Co., Ltd., Tokyo, Japan) and Dormitol (Meiji Co., Ltd., Tokyo, Japan). The DFATs and ASCs were incubated in osteogenic medium for four weeks, and then 1 × 10^7^ cells were seeded on a collagen composite scaffold (AteloCell, KOKEN, Tokyo, Japan). At four weeks, the animals were euthanized with an overdose of isoflurane and implants were removed. All samples were fixed by 4% paraformaldehyde and were decalcified with 10% EDTA. After decalcification, the samples were embedded in paraffin, sectioned at 5 μm in thickness, and stained by hematoxylin-eosin (H&E).

### 4.6. Reverse Transcription-Polymerase Chain Reaction (PCR) Analysis

Total RNA was extracted using TRIzol (Invitrogen). The cDNAs were synthesized with the use of a PrimeScript^®^ II first-strand cDNA synthesis kit (Takara Bio Inc., Otsu, Shiga, Japan) after 10 µg of total RNA obtained from the individual samples were reverse-transcribed separately with Superscript II RT. Briefly, 1 μL of each cDNA was diluted in 25 μL of reaction mixture including 1 × PCR buffer, 1.5 mM MgCl_2_, 200 μM of each dNTP, 0.5 units of Platinum Taq DNA polymerase (TAKARA Bio, Shiga, Japan), and 0.5 μM of each specific primer set.

Primers for RUNX2, osteopontin, DLX5, osterix, and osteocalcin were used as follows: RUNX2: sense 5′-GATGACACTGCCACCTCTGA-3′, antisense 5′-CAGCGTCAACACCATCATTC-3′; osteopontin: sense 5′-TCTGATGAGACCGTCACTGC-3′, antisense 5′-TGTCCTTGTGGCTGTGAAAC-3′; DLX5: sense 5′-TCTCTAGGACTGACGCAAACA-3′, antisense 5′-GTTACACGCCATAGGGTCGC-3′; ssterix: sense 5′-GATAGTGGAGACCTTGCTCGTAG-3′, antisense 5′-GAGGTCACAGGGTATGAGAAGAG-3′; osteocalcin: sense 5′-AGGACCATCTTTCTGCTCACTC, antisense 5′-CTGCCAGAGTTTGGCTTTAG.

### 4.7. Statistical Analysis

All values are the mean ± standard deviation. Statistical analysis in this study was performed using one-way ANOVA and Tukey’s test. *p* < 0.05 was considered significant. All calculations were analyzed using PASW Statistics 18 (SPSS Inc., Chicago, IL, USA).

## 5. Conclusions

In this study, we established a new mesh method to collect DFAT cells. Our results demonstrated that our newly isolated DFAT cells possess osteogenic differentiation potential, and autologous implantation of DFAT cells can contribute to bone regeneration. Our newly established method for isolating DFAT cells may be an attractive source for cell-based bone tissue engineering.

## Figures and Tables

**Figure 1 ijms-22-12392-f001:**
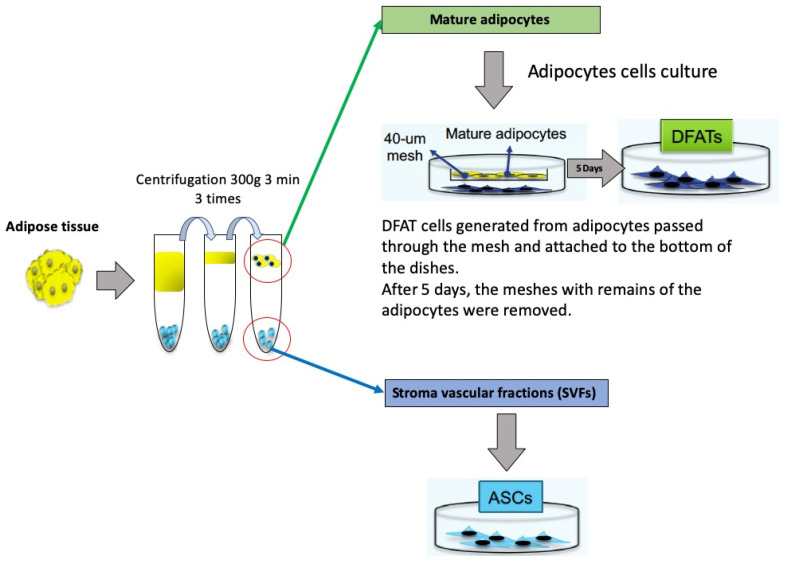
Isolation of DFAT cells and ASCs. The adipose tissue was minced into small pieces and then dissociated in a collagenase and dispase solution. After centrifugation, adipose tissue was separated into two layers. One was the upper layer containing mature adipocytes, and the other was the bottom layer containing cells of the stromal vascular fraction (SVF). Each layer of collected cells was cultured.

**Figure 2 ijms-22-12392-f002:**
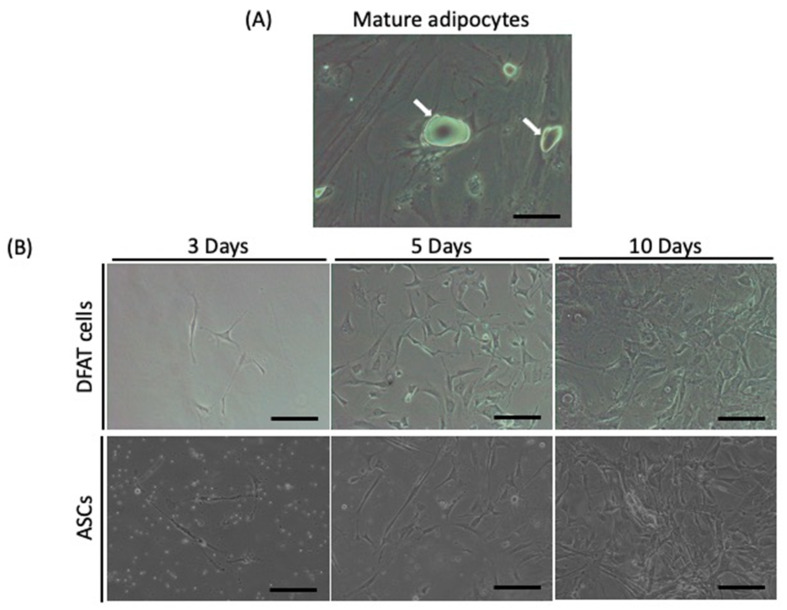
(**A**) Morphology of mature adipocytes. Mature adipocytes had a large lipid droplet (white arrows); (**B**) DFAT cells and ASCs at 3, 5, and 10 days. Both DFAT cells and ASCs showed fibroblast-like shapes. Scale bar: 10 µm.

**Figure 3 ijms-22-12392-f003:**
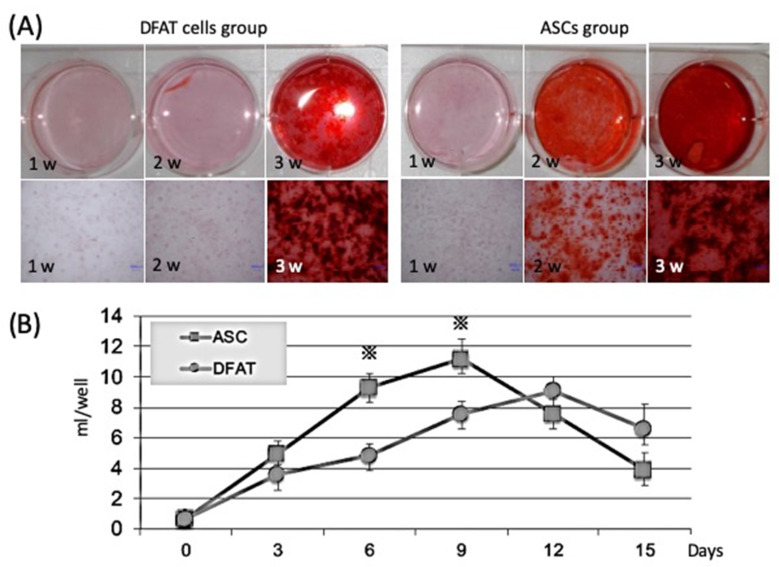
Comparison of osteogenic differentiation capacities by alizarin red staining and ALP activity of DFAT cells and ASCs. (**A**) Mineral depositions of DFAT cells and ASCs were detected by alizarin red staining following 1, 2, and 3 weeks of osteogenic induction. Red areas stained by alizarin red indicate calcium deposition. The well containing DFAT cells cultured in osteogenic medium for 3 weeks and ASCs cultured in osteogenic medium for 2 or 3 weeks were stained by alizarin red, whereas those containing ASCs cultured in osteogenic medium were stained strongly by alizarin red; (**B**) The ALP level of ASCs increased earlier than that of DFAT cells and was significantly different (* *p* < 0.05) on days 6 and 9. The ALP level of DFATs was higher than that of ASCs after day 12.

**Figure 4 ijms-22-12392-f004:**
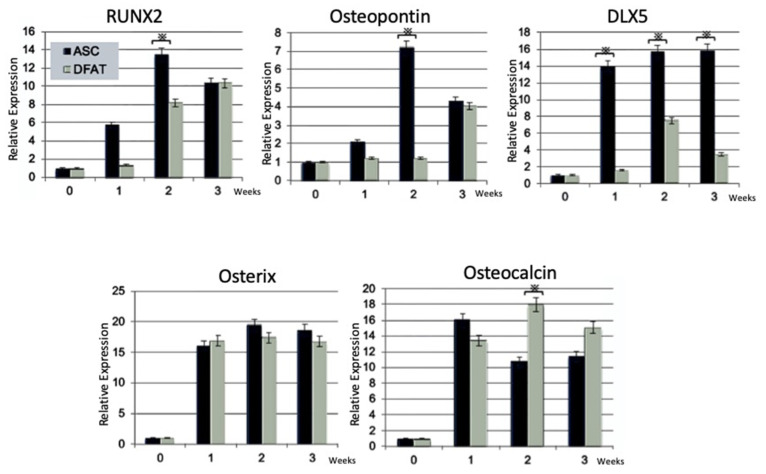
Comparison of osteoblast marker genes (RUNX2, osteopontin, DLX5, osterix, osteocalcin) of DFATs and ASCs as assessed by RT-PCR before (0 weeks) and after osteogenic differentiation culture (1–3 weeks). * *p* < 0.05.

**Figure 5 ijms-22-12392-f005:**
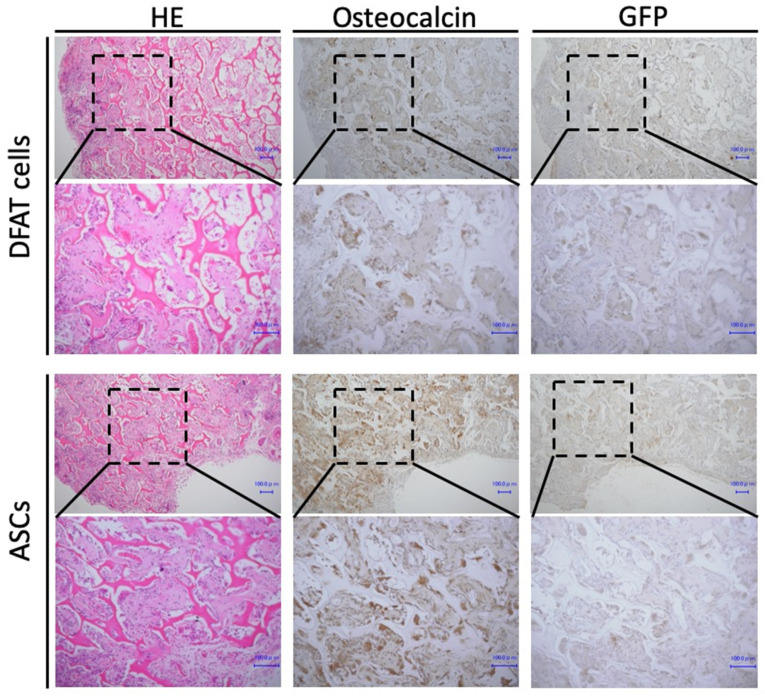
Ectopic bone formation analysis. Hematoxylin and eosin (H&E) stained images of ectopic bone formation at 4 weeks in DFATs group and ASCs group. Scale bar = 100 µm.

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
