# Peer review of "Comparing the Osteogenic Potential and Bone Regeneration Capacities of Dedifferentiated Fat Cells and Adipose-Derived Stem Cells In Vitro and In Vivo: Application of DFAT Cells Isolated by a Mesh Method"

_ijms, 2021, doi:10.3390/ijms222212392_

Round 1
Reviewer 1 Report
In the the conducted research, the authors chose the appropriate methodology allowing for the performance of the intended research. Authors should explain why they did not include the expression results of the surface markers. I think about positive (CD105, CD73, CD90)and negative markers (eg.CD34 and CD45). Have tests been performed to show which cells (adipocytes and chondrocytes) differentiate the isolated stem cells into? Information about such a test should be included.
The conclusions could be more advanced.
Author Response
In the conducted research, the authors chose the appropriate methodology allowing for the performance of the intended research. Authors should explain why they did not include the expression results of the surface markers. I think about positive (CD105, CD73, CD90) and negative markers (eg.CD34 and CD45). Have tests been performed to show which cells (adipocytes and chondrocytes) differentiate the isolated stem cells into? Information about such a test should be included.
The conclusions could be more advanced.
→Thank you for your suggestion. We agree with you. We have performed FACS on adipose stem cells. The results showed CD105, CD73, CD90 positive and CD34 and CD45 negative. However, we have not performed FACS on DFAT cells which were isolated by us because in this study we only focus on the DFAT cells differentiation into bone tissue.
In the future, we will analyze the characteristics of DFAT cells established by the mesh method in detail using FACS. We have added the comment of our study limitation in discussion.
Reviewer 2 Report
The manuscript is hard to read. Many important concepts and findings are hidden because of poor English throughout the entire manuscript. The mesh method here described is interesting and the core of the investigation reported.
Do the authors have any idea about the difference seen in Figure 3. This must be discussed in terms of potential mechanisms of osteogenic differentiation of DFAT.
Author Response
The manuscript is hard to read. Many important concepts and findings are hidden because of poor English throughout the entire manuscript. The mesh method here described is interesting and the core of the investigation reported.
→We have checked and modified the English quality throughout the entire manuscript. This paper has been edited and rewritten by an experienced scientific editor, who has improved the grammar and stylistic expression of the paper.
Do the authors have any idea about the difference seen in Figure 3. This must be discussed in terms of potential mechanisms of osteogenic differentiation of DFAT.
→We have discussed about the difference osteogenic differentiation. We have mentioned that differences between DFAT cells and ASCs in their rate of differentiation into bone tissue and differences in expression markers were thought to be due to the composition of these cell populations. We have modified and added the comments in discussion.

Round 2
Reviewer 1 Report
After the introduced changes, the paper can be published